# Semi-Supervised Neural Architecture Search

[1]**Renqian Luo,**[*] [2]**Xu Tan,** [2]**Rui Wang,** [2]**Tao Qin,** [1]**Enhong Chen,** [2]**Tie-Yan Liu**
[1]University of Science and Technology of China, Hefei, China
[2]Microsoft Research Asia, Beijing, China
[1]lrq@mail.ustc.edu.cn, cheneh@ustc.edu.cn
[2]{xuta, ruiwa, taoqin, tyliu}@microsoft.com

## Abstract

Neural architecture search (NAS) relies on a good controller to generate better architectures or predict the accuracy of given architectures. However, training the controller requires both abundant and high-quality pairs of architectures and their accuracy, while it is costly to evaluate an architecture and obtain its accuracy. In this paper, we propose *SemiNAS*, a semi-supervised NAS approach that leverages numerous unlabeled architectures (without evaluation and thus nearly no cost). Specifically, SemiNAS 1) trains an initial accuracy predictor with a small set of architecture-accuracy data pairs; 2) uses the trained accuracy predictor to predict the accuracy of large amount of architectures (without evaluation); and 3) adds the generated data pairs to the original data to further improve the predictor. The trained accuracy predictor can be applied to various NAS algorithms by predicting the accuracy of candidate architectures for them. SemiNAS has two advantages: 1) It reduces the computational cost under the same accuracy guarantee. On NASBench-101 benchmark dataset, it achieves comparable accuracy with gradient-based method while using only 1/7 architecture-accuracy pairs. 2) It achieves higher accuracy under the same computational cost. It achieves 94.02% test accuracy on NASBench-101, outperforming all the baselines when using the same number of architectures. On ImageNet, it achieves 23.5% top-1 error rate (under 600M FLOPS constraint) using 4 GPU-days for search. We further apply it to LJSpeech text to speech task and it achieves 97% intelligibility rate in the low-resource setting and 15% test error rate in the robustness setting, with 9%, 7% improvements over the baseline respectively.

## 1 Introduction

Neural architecture search (NAS) for automatic architecture design has been successfully applied in several tasks including image classification and language modeling [42, 26, 6]. NAS typically contains two components, a controller (also called generator) that controls the generation of new architectures, and an evaluator that trains candidate architectures and evaluates their accuracy[2]. The controller learns to generate relatively better architectures via a variety of techniques (e.g., reinforcement learning [41, 42], evolution [20], gradient optimization [15, 17], Bayesian optimization [39]), and plays an important role in NAS [41, 42, 18, 20, 17, 15, 39]. To ensure the performance of the controller, a large number of high-quality pairs of architectures and their corresponding accuracy are required as the training data.

---

[*]The work was done when the first author was an intern at Microsoft Research Asia.

[2]Although a variety of metrics including accuracy, model size, and inference speed have been used as search criterion, the accuracy of an architecture is the most important and costly one, and other metrics can be easily calculated with almost zero computation cost. Therefore, we focus on accuracy in this work.

However, collecting such architecture-accuracy pairs is expensive, since it is costly for the evaluator to train each architecture to accurately get its accuracy, which incurs the highest computational cost in NAS. Popular methods usually consume hundreds to thousands of GPU days to discover eventually good architectures [41, 20, 17]. To address this problem, one-shot NAS [2, 18, 15, 35] uses a supernet to include all candidate architectures via weight sharing and trains the supernet to reduce the training time. While greatly reducing the computational cost, the quality of the training data (architectures and their corresponding accuracy) for the controller is degraded [24], and thus these approaches suffer from accuracy decline on downstream tasks.

In various scenarios with limited labeled training data, semi-supervised learning [40] is a popular approach to leverage unlabeled data to boost the training accuracy. In the scenario of NAS, unlabeled architectures can be obtained through random generation, mutation [20], or simply going through the whole search space [32], which incur nearly zero additional cost. Inspired by semi-supervised learning, in this paper, we propose *SemiNAS*, a semi-supervised approach for NAS that leverages a large number of unlabeled architectures. Specifically, SemiNAS 1) trains an initial accuracy predictor with a set of architecture-accuracy data pairs; 2) uses the trained accuracy predictor to predict the accuracy of a large number of unlabeled architectures; and 3) adds the generated architecture-accuracy pairs to the original data to further improve the accuracy predictor. The trained accuracy predictor can be incorporated to various NAS algorithms by predicting the accuracy of unseen architectures.

SemiNAS can be applied to many NAS algorithms. We take the neural architecture optimization (NAO) [17] algorithm as an example, since NAO has the following advantages: 1) it takes architecture-accuracy pairs as training data to train a accuracy predictor to predict the accuracy of architectures, which can directly benefit from SemiNAS; 2) it supports both conventional methods which train each architecture from scratch [42, 20, 17] and one-shot methods which train a supernet with weight sharing [18, 17]; and 3) it is based on gradient optimization which has shown better effectiveness and efficiency. Although we implement SemiNAS on NAO, it is easy to be applied to other NAS methods, such as reinforcement learning based methods [42, 18] and evolutionary algorithm based methods [20].

SemiNAS shows advantages over both conventional NAS and one-shot NAS. Compared to conventional NAS, it can significantly reduce computational cost while achieving similar accuracy, and achieve better accuracy with similar cost. Specifically, on NASBench-101 benchmark, SemiNAS achieves similar accuracy ($93.89\%$) as gradient based methods [17] using only $1/7$ architectures. Meanwhile it achieves $94.02\%$ mean test accuracy surpassing all the baselines when evaluating the same number of architectures (with the same computational cost). Compared to one-shot NAS, SemiNAS achieves higher accuracy using similar computational cost. For image classification, within $4$ GPU days for search, we achieve $23.5\%$ top-1 error rate on ImageNet under the mobile setting. For text to speech (TTS), using $4$ GPU days for search, SemiNAS achieves $97\%$ intelligibility rate in the low-resource setting and $15\%$ sentence error rate in the robustness setting, which outperforms human-designed model by 9 and 7 points respectively. To the best of our knowledge, we are the first to develop NAS algorithms on text to speech (TTS) task. We carefully design the search space and search metric for TTS, and achieve significant improvements compared to human-designed architectures. We believe that our designed search space and metric are helpful for future studies on NAS for TTS.

## 2 Related Work

From the perspective of the computational cost of training candidate architectures, previous works on NAS can be categorized into conventional NAS and one-shot NAS.

Conventional NAS includes [41, 42, 20, 17], which achieve significant improvements on several benchmark datasets. Obtaining the accuracy of the candidate architectures is expensive in conventional NAS, since they train every single architecture from scratch and usually require thousands of architectures to train. The total cost is usually more than hundreds of GPU days [42, 20, 17].

To reduce the huge cost in NAS, one-shot NAS was proposed with the help of weight sharing mechanism. [2] proposes to include all candidate operations in the search space within a supernet and share parameters among candidate architectures. Each candidate architecture is a sub-graph in the supernet and only activates the parameters associated with it. The algorithm trains the supernet and then evaluates the accuracy of candidate architectures by the corresponding sub-graphs in the

supernet. [18, 17, 15, 5, 36, 4, 27, 9] also leverage the one-shot idea to perform efficient search while using different search algorithms. Such weight sharing mechanism successfully cuts down the computational cost to less than 10 GPU days [18, 15, 4, 36]. However, the supernet requires careful design and the training of supernet needs careful tuning. Moreover, it shows inferior performance and reproducibility compared to conventional NAS. One main cause is the short training time and inadequate update of individual architecture [12, 24], which leads to an inaccurate ranking of the architectures, and provides relatively low-quality architecture-accuracy pairs for the controller.

To sum up, there exists a trade-off between computational cost and accuracy. We formalize the computational cost of the evaluator by $C = N \times T$, where $N$ is the number of architecture-accuracy pairs for the controller to learn, and $T$ is the training time of each candidate architecture. In conventional NAS, the evaluator trains each architecture from scratch and the $T$ is typically several epochs[3] to ensure the accuracy of the evaluation, leading to large $C$. In one-shot NAS, the $T$ is reduced to a few mini-batches, which is inadequate for training and therefore produces low-quality architecture-accuracy pairs. Our SemiNAS handles this computation and accuracy trade-off from a new perspective which reduces $N$ by leveraging a large number of unlabeled architectures.

## 3 SemiNAS

In this section, we first describe the semi-supervised training of the accuracy predictor, and then introduce the implementation of the proposed SemiNAS algorithm.

### 3.1 The Semi-Supervised Training of the Accuracy Predictor

To learn from both labeled architecture-accuracy pairs and unlabeled architectures without corresponding accuracy numbers, SemiNAS trains an accuracy predictor via semi-supervised learning. Specifically, we utilize a large number of unevaluated architectures ($M$) to improve the accuracy predictor. To utilize numerous unlabeled data, we leverage self-supervised learning by predicting the accuracy of unevaluated architectures [11] and then combine them with ground-truth data to further improve the accuracy predictor. Following [34], we apply dropout as noise during the training.

However, a simple accuracy predictor is hard to learn information from architectures with pseudo labels via regression task although with techniques in [34]. Inspired by [17], we use an accuracy predictor framework consisting of an encoder $f_e$, a predictor $f_p$ and a decoder $f_d$. The encoder is implemented as an LSTM network to map the discrete architecture $x$ to continuous embedding representations $e_x$, and the predictor uses fully connected layers to predict the accuracy of the architecture taking the continuous embedding $e_x$ as input. The decoder is an LSTM to decode the continuous embedding back to discrete architecture in an auto-regressive manner. The three components are trained jointly via the regression task and the reconstruction task. The semi-supervised learning of the accuracy predictor can be decomposed into 3 steps:

- Train the encoder $f_e$, predictor $f_p$ and the decoder $f_d$ with $N$ architecture-accuracy pairs where each architecture is trained and evaluated.
- Generate $M$ unlabeled architectures and use the trained encoder $f_e$ and predictor $f_p$ to predict their accuracy.
- Use both the $N$ architecture-accuracy pairs and the $M$ self-labeled pairs together to train a better accuracy predictor.

The accuracy predictor learns information from limited number of architecture-accuracy pairs, while there are still numerous unseen architectures. With the help of the decoder, the encoder and the decoder together act like an autoencoder to learn the hidden representation of architectures. Therefore it is able for the whole framework to learn the information of architectures in an unsupervised way without the requirement of ground-truth labels (evaluated accuracy), and further improves the accuracy predictor as the three components are trained jointly. The trained accuracy predictor can be incorporated to various NAS algorithms by predicting the accuracy of unseen architectures for them.

SemiNAS brings advantages over both conventional NAS and one-shot NAS, which can be illustrated under the computational cost formulation $C = N \times T$. Compared to conventional NAS which is

costly, SemiNAS can reduce the computational cost $C$ with smaller $N$ but using more additional unlabeled architectures to avoid accuracy drop, and can also further improve the performance with same computational cost. Compared to one-shot NAS which has inferior accuracy, SemiNAS can improve the accuracy by using more unlabeled architectures under the same computational cost $C$. Specifically, in order to get more accurate evaluation of architectures and improve the quality of architecture-accuracy pairs, we can extend the average training time $T$ for each individual architecture. Meanwhile, we reduce the number of architectures to be trained (i.e., $N$) to keep the total budget $C$ unchanged.

## 3.2  The Implementation of SemiNAS

We now describe the implementation of our SemiNAS algorithm. We take NAO [17] as our implementation since it has following advantages: 1) it contains an encoder-predictor-decoder framework, where the encoder and the predictor can predict the accuracy for large number of architectures without evaluation, and is straightforward to incorporate our method; 2) it performs architecture search by applying gradient ascent which has shown better effectiveness and efficiency; 3) it can incorporate both conventional NAS (whose evaluator trains each architecture from scratch) and one-shot NAS (whose evaluator builds a supernet to train all the architectures via weight sharing).

NAO [17] uses an encoder-predictor-decoder framework as the controller, where the encoder $f_e$ maps the discrete architecture representation $x$ into continuous representation $e_x = f_e(x)$ and uses the predictor $f_p$ to predict its accuracy $\hat{y} = f_p(e_x)$. Then it uses a decoder $f_d$ that is implemented based on a multi-layer LSTM to reconstruct the original discrete architecture from the continuous representation $x = f_d(e_x)$ in an auto-regressive manner.

After the controller is trained, for any given architecture $x$ as the input, NAO moves its representation $e_x$ towards the direction of the gradient ascent of the accuracy prediction $f_p(e_x)$ to get a new and better continuous representation $e'_x$ as follows: $e'_x = e_x + \eta \frac{\partial f_p(e_x)}{\partial e_x}$, where $\eta$ is a step size. $e'_x$ can get higher prediction accuracy $f_p(e'_x)$ after gradient ascent. Then it uses the decoder $f_d$ to decode $e'_x$ into a new architecture $x'$, which is supposed to be better than architecture $x$. The process of the architecture optimization is performed for $L$ iterations, where newly generated architectures at the end of each iteration are added to the architecture pool for evaluation and further used to train the controller in the next iteration. Finally, the best performing architecture in the architecture pool is selected out as the final result.

---

**Algorithm 1** Semi-Supervised Neural Architecture Search

---

1: **Input**: Number of architectures $N$ to evaluate. Number of unlabeled architectures $M$ to use. The set of architecture-accuracy pairs $D = \emptyset$ to train the encoder-predictor-decoder. Number of architectures $K$ based on which to generate better architectures. Training steps $T$ to evaluate each architecture. Number of optimization iterations $L$. Step size $\eta$.
2: Generate $N$ architectures. Use the evaluator to train each architecture for $T$ steps (in conventional way or weight sharing way).
3: Evaluate the $N$ architectures to obtain the accuracy and form the labeled dataset $D$.
4: **for** $l = 1, \cdots, L$ **do**
5:    Train $f_e$, $f_p$ and $f_d$ jointly using $D$.
6:    Randomly generate $M$ architectures and use $f_e$ and $f_p$ to predict their accuracy and forming dataset $\hat{D}$.
7:    Set $\widetilde{D} = D \bigcup \hat{D}$.
8:    Train $f_e$, $f_p$ and $f_d$ using $\widetilde{D}$.
9:    Pick $K$ architectures with top accuracy among $\widetilde{D}$. For each architecture, obtain a better architecture by applying gradient ascent optimization with step size $\eta$.
10:    Evaluate the newly generated architectures using the evaluator and add them to $D$.
11: **end for**
12: **Output**: The architecture in $D$ with the best accuracy.

---

With the semi-supervised method proposed in Section 3.1, we propose our SemiNAS as shown in Alg. 1. First we train the encoder-predictor-decoder on limited number ($N$) of architecture-accuracy pairs (line 5). Then we train the encoder-predictor-decoder with additional $M$ unlabeled architectures (line 6-8). Finally, we perform the step of generating new architectures as the same in [17] (line9-10).

### 3.3 Discussions

Although our SemiNAS is mainly implemented based on NAO in this paper, the key idea of utilizing the trained encoder $f_e$ and predictor $f_p$ to predict the accuracy of numerous unlabeled architectures can be extended to a variety of NAS methods. For reinforcement learning based algorithms [41, 42, 18] where the controller is usually an RNN model, we can predict the accuracy of the architectures generated by the RNN and take the predicted accuracy as the reward to train the controller. For evolution based methods [20], we can predict the accuracy of the architectures generated through mutation and crossover, and then take the predicted accuracy as the fitness of the generated architectures.

## 4 Application to Image Classification

In this section, we demonstrate the effectiveness of SemiNAS on image classification tasks. We first conduct experiments on NASBench-101 [37] and then on the commonly used large-scale ImageNet.

### 4.1 NASBench-101

**Dataset**   NASBench-101 [37] designs a cell-based search space following the common practice [42, 17, 15]. It includes $423, 624$ CNN architectures and trains each architecture CIFAR-10 for 3 times. Querying the accuracy of an architecture from the dataset is equivalent to training and evaluating the architecture. We hope to discover comparable architectures with less computational cost or better architectures with comparable computational cost. Specifically, on this dataset, reducing the computational cost can be regarded as decreasing the number of queries.

**Setup**   Both the encoder and the decoder consist of a single layer LSTM with a hidden size of 16, and the predictor is a three-layer fully connected network with hidden sizes of $16, 64, 1$ respectively. We use Adam optimizer with a learning rate of $0.001$. During search, only valid accuracy is used. After search, we report the mean test accuracy of the selected architecture over the 3 runs. We report two settings of SemiNAS. For the first setting, we use $N = 100, M = 10000$ and up-sample $N$ labeled data by 100x (directly duplicate the labeled data). We generate 100 new architectures based on top $K = 100$ architectures following line 9 in Alg. 1 at each iteration and run for $L = 2$ iterations. The algorithm totally evaluates $100 + 100 \times 2 = 300$ architectures. For the second setting, we set $N = 1100, M = 10000$ and up-sample $N$ labeled data by 10x. We generate 300 new architectures based on top $K = 100$ architectures at each iteration and run for $L = 3$ iterations. The algorithm totally queries $1100 + 300 \times 3 = 2000$ architectures. For comparison, we evaluate random search, regularized evolution (RE) [20] and NAO as baselines, where RE is validated as the best-performing algorithm in the NASBench-101 publication. We limit the number of queries of the baselines to be 2000 for fair comparison. Particularly, we run NAO with two settings using 300 and 2000 architectures for better comparison considering our SemiNAS is mainly implemented based on NAO in this paper. Additionally, we also combine our semi-supervised trained accuracy predictor with RE and name it SemiNAS (RE) for comparison to show the potential of SemiNAS. All the experiments are conducted for 500 times and we report the averaged results. Since the best test accuracy in the dataset is $94.32\%$ and several algorithms are reaching it, we also report test regret (gap to $94.32\%$) following the guide by [37] and the ranking of the accuracy number among the whole dataset to better illustrate the improvements of our method.

**Results**   All the results are listed in Table 1. Random search achieves $93.64\%$ test accuracy with a confidence interval of $[93.61\%, 93.67\%]$ (alpha=99%). This implies that even $0.1\%$ is a significant difference and there exists a large margin for improvement. We can see that, when using the same number of architectures-accuracy pairs (2000), SemiNAS outperforms all the baselines with $94.02\%$ test accuracy and corresponding $0.30\%$ test regret, which ranks top 43 in the whole space. SemiNAS with only 300 architectures achieves $93.89\%$ test accuracy and $0.63\%$ test regret which is on par with NAO with 2000 architectures. Moreover, NAO using 300 architectures only achieves $93.69\%$ which is merely better than random search. This demonstrates that with the help of unlabeled data, SemiNAS indeed outperforms baselines when using the same number of labeled architectures, and can achieve similar performance while using much less resources. Further, SemiNAS (RE) achieves $93.97\%$ which is on par with baseline RE while using only a half number of labeled architectures, and outperforms RE with $94.03\%$ when using same number of labeled architectures (2000). This implies

the potential of using semi-supervised learning in NAS for speeding up the search and applying to various NAS algorithms. We also conduct experiments to study the effect of different number of unlabeled architectures ($M$) and up-sampling ratio in SemiNAS, and the results are in supplementary material.

| Method | #Queries | Test Acc. (%) | SD (%) | Test Regret (%) | Ranking |
|---|---|---|---|---|---|
| Random Search | 2000 | 93.64 | 0.25 | 0.68 | 1749 |
| RE [20] | 2000 | 93.96 | 0.05 | 0.36 | 89 |
| SemiNAS (RE) | 1000 | 93.97 | 0.05 | 0.35 | 76 |
| SemiNAS (RE) | 2000 | 94.03 | 0.05 | 0.29 | 37 |
| NAO [17] | 300 | 93.69 | 0.06 | 0.63 | 1191 |
| NAO [17] | 2000 | 93.90 | 0.03 | 0.42 | 169 |
| SemiNAS | 300 | 93.89 | 0.06 | 0.43 | 197 |
| SemiNAS | 2000 | 94.02 | 0.05 | 0.30 | 43 |

Table 1: Performances of different NAS methods on NASBench-101 dataset. "#Queries" is the number of architecture-accuracy pairs queried from the dataset. "SD" is standard deviation.

## 4.2 ImageNet

Previous experiments on NASBench-101 dataset verify the effectiveness and efficiency of SemiNAS in a well-controlled environment. We further evaluate our approach to the large-scale ImageNet dataset.

**Search space**  We adopt a MobileNet-v2 [23] based search space following ProxylessNAS [4]. It consists of multiple stacked layers. We search the operation of each layer. Candidate operations include mobile inverted bottleneck convolution layers [23] with various kernel sizes $\{3, 5, 7\}$ and expansion ratios $\{3, 6\}$, as well as zero-out layer.

**Setup**  We randomly sample $50,000$ images from the training data as valid set for architecture search. Since training ImageNet is too expensive, we adopt weight sharing mechanism [18, 4] to perform one-shot search. We train the supernet on $4$ GPUs for 20000 steps with a batch size of 128 per card. We set $N = 100, M = 4000$ and run the search process for $L = 3$ iterations. In each iteration, 100 new better architectures are generated based on top $K = 100$ architectures following line 9 in Alg. 1. The search runs for 1 day on 4 V100 GPUs. To fairly compare with other works, we limit the FLOPS of the discovered architecture to be less than 600M. The discovered architecture is trained for 300 epochs with a total batch size of 256. We use the SGD optimizer with an initial learning rate of 0.05 and a cosine learning rate schedule [16]. More training details are in the supplementary details. For NAO, we use the open source code [4] and train it on the same search space used in this paper. In both SemiNAS and NAO, we train the supernet for 20000 steps at each iteration to keep the same cost while NAO uses larger $N = 1000$. For ProxylessNAS, since it also optimizes latency as additional target, for fair comparison, we use their open source code [5] and rerun the search while optimizing accuracy without considering latency. We limit the FLOPS of discovered architecture to be less than 600M. We run all the experiments for 5 times.

**Results**  From the results in Table 2, SemiNAS achieves $23.5\%$ top-1 test error rate on ImageNet under the 600M FLOPS constraint, which outperforms all the other NAS works. Specifically, it significantly **outperforms the baseline algorithm NAO based on which SemiNAS is mainly implemented by** $1.0\%$, and outperforms ProxylessNAS where our search space is based on by $0.5\%$.

## 5  Application to Text to Speech

In this section, we further explore the application of SemiNAS to a new task: text to speech.

| Model/Method | Top-1 (%) | Top-5 (%) | Params (Million) | FLOPS (Million) |
|---|---|---|---|---|
| MobileNetV2 [23] | 25.3 | - | 6.9 | 585 |
| ShuffleNet $2\times$ (v2) [38] | 25.1 | - | $\sim 5$ | 591 |
| NASNet-A [41] | 26.0 | 8.4 | 5.3 | 564 |
| AmoebaNet-A [20] | 25.5 | 8.0 | 5.1 | 555 |
| PNAS [14] | 25.8 | 8.1 | 5.1 | 588 |
| SNAS [35] | 27.3 | 9.2 | 4.3 | 522 |
| DARTS [15] | 26.9 | 9.0 | 4.9 | 595 |
| P-DARTS [5] | 24.4 | 7.4 | 4.9 | 557 |
| PC-DARTS [36] | 24.2 | 7.3 | 5.3 | 597 |
| Efficienet-B0 [29] | 23.7 | 6.8 | 5.3 | 390 |
| Random Search | 25.2 | 8.0 | 5.1 | 578 |
| ProxylessNAS [4] | 24.0 | 7.1 | 5.8 | 595 |
| NAO [17] | 24.5 | 7.5 | 6.5 | 590 |
| SemiNAS | **23.5** | **6.8** | 6.3 | 599 |

Table 2: Performances of different methods on ImageNet. For fair comparison, we run NAO on the same search space used in this paper, and run ProxylessNAS by optimizing accuracy without latency.

Text to speech (TTS) [31, 25, 19, 13, 21] is an import task aiming to synthesize intelligible and natural speech from text. The encoder-decoder based neural TTS [25] has achieved significant improvements. However, due to the different modalities between the input (text) and the output (speech), popular TTS models are still complicated and require many human experiences when designing the model architecture. Moreover, unlike many other sequence learning tasks (e.g., neural machine translation) where the Transformer model [30] is the dominate architecture, RNN based Tacotron [31, 25], CNN based Deep Voice [1, 7, 19], and Transformer based models [13] show comparable accuracy in TTS, without one being exclusively better than others.

The complexity of the model architecture in TTS indicates great potential of NAS on this task. However, applying NAS on TTS task also has challenges, mainly in two aspects: 1) Current TTS model architectures are complicated, including many human designed components. It is difficult but important to design the network bone and the corresponding search space for NAS. 2) Unlike other tasks (e.g., image classification) whose evaluation is objective and automatic, the evaluation of a TTS model requires subject judgement and human evaluation in the loop (e.g., intelligibility rate for understandability and mean opinion score for naturalness). It is impractical to use human evaluation for thousands of architectures in NAS. Thus, it is difficult but also important to design a specific and appropriate objective metric as the reward of an architecture during the search process. Next, we design the search space and evaluation metric for NAS on TTS, and apply SemiNAS on two specific TTS settings: low-resource setting and robustness setting.

## 5.1 Experiment Settings

**Search space**   After surveying the previous neural TTS models, we choose a multi-layer encoder-decoder based network as the network backbone for TTS. We search the operation of each layer of the encoder and the decoder. The search space includes 11 candidate operations in total: convolution layer with kernel size $\{1, 5, 9, 13, 17, 21, 25\}$, Transformer layer [13] with number of heads of $\{2, 4, 8\}$ and LSTM layer. Specifically, we use unidirectional LSTM layer, causal convolution layer, causal self-attention layer in the decoder to avoid seeing the information in future positions. Besides, every decoder layer is inserted with an additional encoder-decoder-attention layer to catch the relationship between the source and target sequence, where the dot-product multi-head attention in Transformer [30] is adopted.

**Evaluation metric**   It has been shown that the quality of the attention alignment between the encoder and decoder is an important influence factor on the quality of synthesized speech in previous works [21, 31, 25, 13, 19], and misalignment can be observed for most mistakes (e.g., skipping and repeating). Accordingly, we consider the diagonal focus rate (DFR) of the attention map between the encoder and decoder as the metric of an architecture. DFR is defined as: $DFR = \frac{\sum_{i=1}^{I} \sum_{o=ki-b}^{ki+b} A_{o,i}}{\sum_{i=1}^{I} \sum_{o=1}^{O} A_{o,i}}$,

where $A \in \mathbb{R}^{O \times I}$ denotes the attention map, $I$ and $O$ are the length of the source input sequence and the target output sequence, $k = \frac{O}{I}$ is the slope factor and $b$ is the width of the diagonal area in the attention map. DFR measures how much attention lies in the diagonal area with width $b$ in the attention matrix, and ranges in $[0, 1]$ which is the larger the better. In addition, we have also tried valid loss as the search metric, but it is inferior to DFR according to our preliminary experiments.

**Task setting**    Current TTS systems are capable of achieving near human-parity quality when trained on adequate data and tested on regular sentences [25, 13]. However, current TTS models have poor performance on two specific TTS settings: 1) low-resource setting, where only few paired speech and text data is available. 2) Robustness setting, where the test sentences are not regular (e.g., too short, too long, or contain many word pieces that have the same pronunciations). Under these two settings, the synthesized speech of a human-designed TTS model is usually not accurate and robust (i.e., some words are skipped or repeated). Thus we apply SemiNAS on these two settings to improve the accuracy and robustness. We conduct experiments on the LJSpeech dataset [10] which contains 13100 text and speech data pairs with approximately 24 hours of speech audio.

## 5.2    Results on Low-Resource Setting

**Setup**    To simulate the low-resource scenario, we randomly split out 1500 paired speech and text samples as the training set, where the total audio length is less than 3 hours. We use $N = 100, M = 4000, T = 3$. We adopt the weight sharing mechanism and train the supernet on 4 GPUs for 1000 epochs. The search runs for 1 day on 4 P40 GPUs. Besides, we train vanilla NAO as a baseline where $N = 1000$. The discovered architecture is trained on the training set for 80k steps on 4 GPUs, with a batch size of 30K speech frames on each GPU. More details are provided in the supplementary materials. In the inference process, the output mel-spectrograms are transformed into audio samples using Griffin-Lim [8]. We run all the experiments for 5 times.

| Model/Method | Intelligibility Rate (%) | DFR (%) |
|---|---|---|
| Transformer TTS [13] | 88 | 86 |
| NAO [17] | 94 | 88 |
| SemiNAS | **97** | **90** |

Table 3: Results on LJSpeech under the low-resource setting. "DFR" is diagonal focus rate.

**Results**    We test the performance of SemiNAS, NAO [17] and Transformer TTS (following [13]) on the 100 test sentences and report the results in Table 3. We measure the performances in terms of word level intelligibility rate (IR), which is a commonly used metric to evaluate the quality of generated audio [22]. IR is defined as the percentage of test words whose pronunciation is considered to be correct and clear by human. It is shown that SemiNAS achieves 97% IR, with significant improvements of 9 points over human designed Transformer TTS and 3 points over NAO. We also list the DFR metric for each method in Table 3, where SemiNAS outperforms Transformer TTS and NAO in terms of DFR, which is consistent with the results on IR and indicates that our proposed search metric DFR can indeed guide NAS algorithms to achieve better accuracy. We also use MOS (mean opinion score) [28] to evaluate the naturalness of the synthesized speech. Using Griffin-Lim as the vocoder to synthesize the speech, the ground-truth mel-spectrograms achieves 3.26 MOS, Transformer TTS achieves 2.25, NAO achieves 2.60 and SemiNAS achieves 2.66. SemiNAS outperforms other methods in terms of MOS, which also demonstrates the advantages of SemiNAS. We also attach the discovered architecture by SemiNAS in the supplementary materials.

## 5.3    Results on Robustness Setting

**Setup**    We train on the whole LJSpeech dataset as the training data. For robustness test, we select the 100 sentences as used in [19] (attached in the supplementary materials) that are found hard for TTS models. Training details follow the same as in the low-resource TTS experiment. We also attach the discovered architecture in the supplementary materials. We run all the experiments for 5 times.

**Results**    We report the results in Table 4, including the DFR, the number of sentences with repeating and skipping words, and the sentence level error rate. A sentence is counted as an error if it contains

| Model/Method | DFR (%) | Repeat | Skip | Error (%) |
|---|---|---|---|---|
| Transformer TTS[13] | 15 | 1 | 21 | 22 |
| NAO [17] | 25 | 2 | 18 | 19 |
| SemiNAS | **30** | 2 | 14 | **15** |

Table 4: Robustness test on the 100 hard sentences. "DFR" stands for diagonal focus rate.

a repeating or skipping word. SemiNAS is better than Transformer TTS [13] and NAO [17] on all the metrics. It reduces the error rate by 7% and 4% compared to Transformer TTS structure designed by human experts and the searched architecture by NAO respectively.

## 6   Conclusion

High-quality architecture-accuracy pairs are critical to NAS; however, accurately evaluating the accuracy of an architecture is costly. In this paper, we proposed SemiNAS, a semi-supervised learning method for NAS. It leverages a small set of high-quality architecture-accuracy pairs to train an initial accuracy predictor, and then utilizes a large number of unlabeled architectures to further improve the accuracy predictor. Experiments on image classification tasks (NASBench-101 and ImageNet) and text to speech tasks (the low-resource setting and robustness setting) demonstrate 1) the efficiency of SemiNAS on reducing the computation cost over conventional NAS while achieving similar accuracy and 2) its effectiveness on improving the accuracy of both conventional NAS and one-shot NAS under similar computational cost. In the future, we will apply SemiNAS to more tasks such as automatic speech recognition, text summarization, etc. Furthermore, we will explore advanced semi-supervised learning methods [33, 3] to improve SemiNAS.

## Broader Impact

This work focuses on neural architecture search. It has the following potential positive impact in the society: 1) Improve the performance of neural networks for better applications. 2) Reduce the human efforts in designing neural architectures. At the same time, it may have some negative consequences because architecture search may cost many resources.

## Funding Disclosure

Funding in direct support of this work: computation resources provided by Microsoft.

## Footnotes

[3]One epoch means training on the whole dataset for once.

[4]`https://github.com/renqianluo/NAO_pytorch`

[5]`https://github.com/mit-han-lab/proxylessnas`

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
