[Supplementary Material]

# Appendix for Paper: Semi-Supervised Neural Architecture Search

[1]**Renqian Luo,**[*][2]**Xu Tan,** [2]**Rui Wang,** [2]**Tao Qin,** [1]**Enhong Chen,** [2]**Tie-Yan Liu**
[1]University of Science and Technology of China, Hefei, China
[2]Microsoft Research Asia, Beijing, China
[1]lrq@mail.ustc.edu.cn, cheneh@ustc.edu.cn
[2]{xuta, ruiwa, taoqin, tyliu}@microsoft.com

## 1 Experiment Details

### 1.1 NASBench-101

In the first setting where $N = 100, M = 10000$, 100 new architectures are generated based on top $K = 100$ architectures at each iteration. In the second setting where $N = 1100, M = 10000$, 300 new architectures are generated based on top $K = 100$ architectures at each iteration. We use $\lambda = 0.8$ as the trade-off parameter to balance the regression loss and the reconstruction loss.

### 1.2 ImageNet

We build the supernet following [1]. We train the supernet on 4 GPUs for 20000 steps with a batch size of 128 per card. We use SGD optimizer with a learning rate of 1.6 and decay the learning rate by a factor of 0.97 per epoch. The discovered architecture is trained on 4 P40 cards for 300 epochs with a batch size of 64 per card. We use the SGD optimizer with an initial learning rate of 0.05 and a cosine learning rate schedule [4].

### 1.3 TTS

We adopt the weight sharing mechanism for search and train the supernet on 4 GPUs. The discovered architecture is trained on the training set for 80k steps on 4 GPUs, with a batch size of 30K speech frames on each GPU. We use the Adam optimizer with $\beta_1 = 0.9, \beta_2 = 0.98, \epsilon = 1e - 9$ and follow the same learning rate schedule in [3] with 4000 warmup steps.

## 2 Study of SemiNAS

In this section, we conduct experiments on NASBench-101 to study SemiNAS, including the number of unlabeled architectures $M$ and the up-sampling ratio of labeled architectures.

**Number of unlabeled architectures** $M$  We study the effect of different $M$ on SemiNAS. Given $N = 100$, we range $M$ within $\{0, 100, 200, 500, 1000, 2000, 5000, 10000\}$, and plot the results in Fig. 1(a). Notice that $M = 0$ is equivalent to NAO without using any additional evaluated architectures. We can see that the test accuracy increases as $M$ increases, indicating that utilizing unlabeled architectures indeed helps the training of the controller and generating better architectures.

**Up-sampling ratio**  Since $N$ is much smaller than $M$, we do up-sampling to balance the data. We study how the up-sampling ratio affects the effectiveness of SemiNAS on NASBench-101. We set

---

[*]The work was done when the first author was an intern at Microsoft Research Asia.

Figure 1: Study of SemiNAS on NASBench-101. (a): Performances with different $M$. (b): Performances with different up-sampling ratios.

$N = 100$, $M = 10000$ and range the up-sampling ratio in $\{1, 2, 5, 10, 20, 50, 100\}$ where 1 means no up-sampling. The results are depicted in Figure 1(b). We can see that the final accuracy would benefit from up-sampling but will not continue to improve when the ratio is high (e.g., larger than 10).

## 3 Discovered Architectures

We show the discovered architectures for the tasks by SemiNAS.

### 3.1 ImageNet

We adopt the ProxylessNAS [1] search space which is built on the MobileNet-V2 [7] backbone. It contains several different stages and each stage consists of multiple layers. We search the operation of each individual layer. There are 7 candidate operations in the search space:

- MBConv (k=3, r=3)
- MBConv (k=3, r=6)
- MBConv (k=5, r=3)
- MBConv (k=5, r=6)
- MBConv (k=7, r=3)
- MBConv (k=7, r=6)
- zero-out layer

where MBConv is mobile inverted bottleneck convolution, k is the kernel size and r is the expansion ratio [7]. Our discovered architecture for ImageNet is depicted in Fig. 2

### 3.2 TTS

We adopt encoder-decoder based architecture as the backbone, and search the operation of each layer. Candidate operations include:

- Convolution layer with kernel size of 1
- Convolution layer with kernel size of 5
- Convolution layer with kernel size of 9
- Convolution layer with kernel size of 13
- Convolution layer with kernel size of 17
- Convolution layer with kernel size of 21

Figure 2: Architecture for ImageNet discovered by SemiNAS. "MBConv3" and "MBConv6" denote mobile inverted bottleneck convolution layer with an expansion ratio of 3 and 6 respectively.

- Convolution layer with kernel size of 25
- Transformer layer with head number of 2
- Transformer layer with head number of 4
- Transformer layer with head number of 8
- LSTM layer

### 3.2.1 Low-Resource Setting

The discovered architecture by SemiNAS for low-resource setting is shown in Fig. 3

Figure 3: Architecture for low-resource setting discovered by SemiNAS.

### 3.2.2 Robustness Setting

The discovered architecture by SemiNAS for robustness setting is shown in Fig. 4

Figure 4: Architecture for robustness setting discovered by SemiNAS.

# 4   Robustness Test Sentences

We list the 100 sentences we use for robustness setting:
a b c.
x y z.
hurry.
warehouse.
referendum.
is it free?
justifiable.
environment.
a debt runs.
gravitational.
cardboard film.
person thinking.
prepared killer.
aircraft torture.
allergic trouser.
strategic conduct.
worrying literature.
christmas is coming.
a pet dilemma thinks.
how was the math test?
good to the last drop.
an m b a agent listens.
a compromise disappears.
an axis of x y or z freezers.
she did her best to help him.
a backbone contests the chaos.
two a greater than two n nine.
don't step on the broken glass.
a damned flips into the patient.

a trade purges within the b b c.
i'd rather be a bird than a fish.
i hear that nancy is very pretty.
i want more detailed information.
please wait outside of the house.
n a s a exposure tunes the waffle.
a mist dictates within the monster.
a sketch ropes the middle ceremony.
every farewell explodes the career.
she folded here handkerchief neatly.
against the steam chooses the studio.
rock music approaches at high velocity.
nine adam baye study on the two pieces.
an unfriendly decay conveys the outcome.
abstraction is often one floor above you.
a played lady ranks any publicized preview.
he told us a very exciting adventure story.
on august twenty eight mary plays the piano.
into a controller beams a concrete terrorist.
i often see the time eleven eleven on clocks.
it was getting dark and we weren't there yet.
against every rhyme starves a choral apparatus.
everyone was busy so i went to the movie alone.
i checked to make sure that he was still alive.
a dominant vegetarian shies away from the g o p.
joe made the sugar cookies susan decorated them.
i want to buy a onesie but know it won't suit me.
a former override of q w e r t y outside the pope.
f b i says that c i a says i'll stay way from it.
any climbing dish listens to a cumbersome formula.
she wrote him a long letter but he didn't read it.
dear beauty is in the heat not physical i love you.
an appeal on january fifth duplicates a sharp queen.
a farewell solos on march twenty third shakes north.
he ran out of money so he had to stop playing poker.
for example a newspaper has only regional distribution t.
i currently have four windows open up and i don't know why.
next to my indirect vocal declines every unbearable academic.
opposite her sounding bag is a m c's configured thoroughfare.
from april eighth to the present i only smoke four cigarettes.
i will never be this young again every oh damn i just got older.
a generous continuum of amazon dot com is the conflicting worker.
she advised him to come back at once the wife lectures the blast.
a song can make or ruin a person's day if they let it get to them.
she did not cheat on the test for it was not the right thing to do.
he said he was not there yesterday however many people saw him there.
should we start class now or should we wait for everyone to get here?
if purple people eaters are real where do they find purple people to eat?
on november eighteenth eighteen twenty one a glittering gem is not enough.
a rocket from space x interacts with the individual beneath the soft flaw.
malls are great places to shop i can find everything i need under one roof.
i think i will buy the red car or i will lease the blue one the faith nests.
italy is my favorite country in fact i plan to spend two weeks there next year.
i would have gotten w w w w dot google dot com but my attendance wasn't good enough.
nineteen twenty is when we are unique together until we realise we are all the same.
my mum tries to be cool by saying h t t p colon slash slash w w w b a i d u dot com.
he turned in the research paper on friday otherwise he emailed a s d f at yahoo dot org.
she works two jobs to make ends meet at least that was her reason for no having time to join us.
a remarkable well promotes the alphabet into the adjusted luck the dress dodges across my assault.

a b c d e f g h i j k l m n o p q r s t u v w x y z one two three four five six seven eight nine ten.

across the waste persists the wrong pacifier the washed passenger parades under the incorrect computer.

if the easter bunny and the tooth fairy had babies would they take your teeth and leave chocolate for you?

sometimes all you need to do is completely make an ass of yourself and laugh it off to realise that life isn't so bad after all.

she borrowed the book from him many years ago and hasn't yet returned it why won't the distinguishing love jump with the juvenile?

last friday in three week's time i saw a spotted striped blue worm shake hands with a legless lizard the lake is a long way from here.

i was very proud of my nickname throughout high school but today i couldn't be any different to what my nickname was the metal lusts the ranging captain charters the link.

i am happy to take your donation any amount will be greatly appreciated the waves were crashing on the shore it was a lovely sight the paradox sticks this bowl on top of a spontaneous tea.

a purple pig and a green donkey flew a kite in the middle of the night and ended up sunburn the contained error poses as a logical target the divorce attacks near a missing doom the opera fines the daily examiner into a murderer.

as the most famous singer-songwriter jay chou gave a perfect performance in beijing on may twenty fourth twenty fifth and twenty sixth twenty three all the fans thought highly of him and took pride in him all the tickets were sold out.

if you like tuna and tomato sauce try combining the two it's really not as bad as it sounds the body may perhaps compensates for the loss of a true metaphysics the clock within this blog and the clock on my laptop are on hour different from each other.

someone i know recently combined maple syrup and buttered popcorn thinking it would taste like caramel popcorn it didn't and they don't recommend anyone else do it either the gentleman marches around the principal the divorce attacks near a missing doom the color misprints a circular worry across the controversy.

## 5 Demo of TTS

We provide demo for both low-resource setting and robustness setting of TTS experiments. Specifically, we provide 10 test cases for each setting respectively and provide their ground-truth audio (if exist), generated audio by Transformer TTS and generated audio by SemiNAS. All can be found in the folder **tts_demo**. In the folder, **low_resource** and **reobustness** represent the low-resource setting and the robustness setting. In each of these two folders, **test_text.txt** contains the 10 test cases from the test set. Folder **GriffinLim** contains the audio synthesized by GriffinLim [2] using the ground-truth mel spectrogram. Folder **TransformerTTS** contains the audio synthesized by GriffinLim using the mel spectrogram generated by Transformer TTS [3]. Folder **SemiNAS** contains the audio synthesized by GriffinLim using the mel spectrogrtam generated by the architecture discovered by SemiNAS. Note that there is no ground-truth audio for robustness setting.

## 6 Implementation Details

We implement all the code in Pytorch [6] with version 1.2. We implement the core architecture search algorithm following NAO [5][2]. For downstream tasks, we implement the code following corresponding baselines. For ImageNet experiment, we build our code based on ProxylessNAS implementation [3]. For TTS experiment, we build the code following Transformer TTS [3] which is originally in Tensorflow.

## Footnotes

[2]https://github.com/renqianluo/NAO_pytorch

[3]https://github.com/mit-han-lab/proxylessnas

[2] Daniel Griffin and Jae Lim. Signal estimation from modified short-time fourier transform. *IEEE Transactions on Acoustics, Speech, and Signal Processing*, 32(2):236–243, 1984.

[3] Naihan Li, Shujie Liu, Yanqing Liu, Sheng Zhao, and Ming Liu. Neural speech synthesis with transformer network. In *Proceedings of the AAAI Conference on Artificial Intelligence*, volume 33, pages 6706–6713, 2019.

[4] Ilya Loshchilov and Frank Hutter. Sgdr: Stochastic gradient descent with warm restarts. *arXiv preprint arXiv:1608.03983*, 2016.

[5] Renqian Luo, Fei Tian, Tao Qin, and Tie-Yan Liu. Neural architecture optimization. *arXiv preprint arXiv:1808.07233*, 2018.

[6] Adam Paszke, Sam Gross, Francisco Massa, Adam Lerer, James Bradbury, Gregory Chanan, Trevor Killeen, Zeming Lin, Natalia Gimelshein, Luca Antiga, et al. Pytorch: An imperative style, high-performance deep learning library. In *Advances in neural information processing systems*, pages 8026–8037, 2019.

[7] Mark Sandler, Andrew Howard, Menglong Zhu, Andrey Zhmoginov, and Liang-Chieh Chen. Mobilenetv2: Inverted residuals and linear bottlenecks. In *Proceedings of the IEEE Conference on Computer Vision and Pattern Recognition*, pages 4510–4520, 2018.

## References

[1] Han Cai, Ligeng Zhu, and Song Han. Proxylessnas: Direct neural architecture search on target task and hardware. *arXiv preprint arXiv:1812.00332*, 2018.