[Reviews · NeurIPS 2020]

Review 1

Summary and Contributions: - The authors propose a semi-supervised learning approach to architecture search. They first train a network to predict the accuracy on a small amount of labeled architecture-accuracy pairs. They then use this trained network to label the accuracy of lots of architectures that they have never trained on before. - All of these labeled and pseudo-labeled architectures are then added together and trained jointly to improve the performance of the controller network, while incurring minimal additional cost. - Their method can work with many different NAS algorithms, but for most results they build off of the NAO [15] algorithm. - They yield strong results on three different tasks/search spaces by both improving the previous best results and getting large efficiency improvements.

Strengths: - The authors propose an interesting idea for using a standard semi-supervised pseudo labeling approach to improve the efficiency of neural architecture search - The authors run their method on a variety of different tasks/search spaces: Text to Speech, NasBench-101 and ImageNet - They make sure to run their closest baseline (NAO) with the same setup to be sure the comparisons are fair - The authors opensource code for reproducibility - The results achieved on all three setups are solid (improving both efficiency and best result), with their method not adding much complexity to the algorithm or the training time

Weaknesses: - The authors did not go much into why the semi-supervised learning method appears to help. In most pseudo labeling methods in other domains, training with noise (augmentation, dropout, etc...) appears to be quite important for the method to work well (e.g. Self-training with Noisy Student improves ImageNet classification by Xie et al.). When re-training the controller on the labeled and pseudo-labeled architecture-accuracy pairs, there is no new information being added to the system without noise being used. A section talking about this would be informative to the reader on why this semi-supervised method was improving performance. - In table 1, it would be nice to see how the method helps in an even lower query regime like 50. - Additionally, the baselines in Table 2 are not very strong and most have been published before 2018.

Correctness: Yes the claims, method and its empirical methodologies are correct.

Clarity: Yes the paper is clearly written and easy to understand

Relation to Prior Work: Yes the authors do a good job of explaining the prior work and how they build on top of it

Reproducibility: Yes

Additional Feedback:


Review 2

Summary and Contributions: This paper introduces semi-supervised learning on top of nerual architecture optimization, which consists of a network performance predictor. It samples a random set of architectures from the search space, evaluate and obtain the ground-truth performance and to improve the training of performance prediction. Experiments are conducted on ImageNet for image classification and LJSpeech for neural language processing tasks. This method is also validated on benchmark search space of NASBench-101.

Strengths: Applying semi-supervised learning into NAS is reasonable. Since the current regime of weight sharing NAS is flawed, using some ground-truth label to guide the search of sampling algorithm is a interesting idea. The cost of SemiNAS is in between the one-shot NAS and full NAS approach, thus providing the practitioner in NAS community a choice based on their budget. Experiments across classification and NLP are solid. Experiments on NASBench-101 also shows the semi-nas can discover better architecture in terms of the ranking. On large scale datasets, it also clearly improve the original NAO by a big margin without introducing extra FLOPS or parameters.

Weaknesses: Authors should tune down their Semi-NAS, rather it is more appropriate to name this method Semi-NAO as this is the only method it applies to. Otherwise, the title seems overclaming the content that is not covered in this paper. And we will not know if the Semi-NAS can really work on other algorithms based on the current version. I look forwards to hearing back from the authors to decide my final rating.

Correctness: Yes, the method is correct both for the derivation and evaluation.

Clarity: It is well written and easy to understand.

Relation to Prior Work: The relation comparing to previous work is included in the related work section.

Reproducibility: Yes

Additional Feedback: --- Post rebuttal comments --- The rebuttal clears my concern, thus I raise the score to acceptance.


Review 3

Summary and Contributions: The paper proposed a semi-supervised NAS approach that can leverage unlabeled architectures. More particularly, this work applies the NAO approach that conducts a search on the continuous latent space using a cost model trained with labeled and unlabeled data.

Strengths: [Novelty on using SSL] Applying SSL to train an accuracy predictor is an interesting direction. This paper provides a simple but effective solution of semi-supervised learn a cost model to augment the training data for a NAS controller. The reviewer likes the simplicity of the algorithm. The paper also demonstrates stronger empirical results than a baseline NAO approach and many related approaches. The ImageNet result is 1% better than NAO, which is significant. The paper also explains the experimental setting and training cost much better than a previous version. The evaluation on ImageNet and on a speech workload (particularly in a low-resource setting) are interesting.

Weaknesses: [Empirical results are not particularly strong] In Table 1, second block SemiNAS (RE) barely outperforms RE. Also in Table 1, third block, SemiNAS outperforms NAO only by 0.12%. While these numbers come from NASBench-101, we all know that empirically, CIFAR-10 accuracy of *any* models and experiments have a standard deviation of about 0.05 - 0.1, and hence I would say there is no improvement between SemiNAS and other baselines. Meanwhile, Table 2 also includes some irrelevant baselines, namely those in the second block, while omits an important baseline. EfficientNet-B0 (Tan and Le, 2019) was found in the same search space of MobileNet-V2, has 390M FLOPs, and has the ImageNet top-1 error of 23.2, which is about the same as 23.5 of SemiNAS. Why is this not reported? Note that EfficientNet-B0 is found using vanilla NAS, which is way more expensive than SemiNAS, and so such comparison won’t devalue SemiNAS. [The selection of search method is not intuitive] The underlying search infrastructure is built on top of NAO, which is an encoder-predictor-decoder framework. However, it is a less common search, as compared to a more common end-to-end search, or a differentiable search, or a one-shot search. The reviewer would like to understand the incentives of selecting a less commonly used search infrastructure. [Novelty not strong] Semi-supervised learning applied in NAS is not entirely new. Pseudo labels [3] and consistency regularization [4] are well-established methods. The paper should at least compare with those classic papers. The paper also does not compare to important relate work [1-2], applying SSL in NAS. [1] Qizhe Xie, and others, "Unsupervised Data Augmentation for Consistency Training" [2] Hieu Pham, Quoc V. Le, “Semi-supervised Learning by Coaching” [3] Dong-Hyun Lee, “Pseudo-Label : The Simple and Efficient Semi-Supervised Learning Method for Deep Neural Networks” [4] David Berthelot, “MixMatch: A Holistic Approach to Semi-Supervised Learning”

Correctness: [Incorrect usage of terminologies] Section 3.1 is titled “Semi-supervised training of the controller”, and I think this is inaccurate. Technically, the controller of SemiNAS is still trained with reinforcement learning, where the reward is given by the accuracy predictor’s output.

Clarity: [Convoluted description of how the predictor is trained] The paragraph at Lines 130-141 is very confusing. My understanding of the controller / predictor relationship is as follows: 1. A controller can generate an architecture x, which is embedding as e_x 2. An LSTM encoder reads e_x, passes the resulting hidden representation to f_p to predict x’s accuracy. This f_p could be trained using a supervised regression loss. 3. An LSTM decoder starts from the encoded e_x, passes e_x’s representation to f_d, which reconstructs x by decoding an auto-regressive sequence. The reconstruction loss does not depend on any label. Thus, the final training objective is semi-supervised (supervised in (2), unsupervised in (3)). Now, I COULD BE COMPLETELY WRONG in my understanding above, and that is the point! The presentation here is very convoluted. The authors could have (and should have) used a figure and a few equations to make clear of these points. Even if I understood the method correctly, much details remains unclear, such as:

Relation to Prior Work: The paper can be improved by comparing to more related work in SSL and applying SSL in NAS. The paper also should differentiate the proposed approach with related work better. The reviewer likes the thoroughness of the paper, especially in its evaluation. However, the novelty side should be better addressed.

Reproducibility: Yes

Additional Feedback: Additional question: - How are the controller / predictor / encoder / decoder weights organized? - Do the controller and the encoder use the same weights? - How is f_p parameterized? For an improved version, the reviewer recommend differentiating the proposed method with related work in SSL, such as pseudo labeling, teacher-student, consistency regularization, etc. What makes it more challenging when applying the SSL method to a NAS problem? For more conclusive results, what is the top-1 accuracy on ImageNet if we use the same amount of labeled data while being able to generate unlimited unlabeled data to train the controller network? Also, to achieve the same ImageNet accuracy, what is the minimum amount of labeled data does this approach take? How does it compare with a related work like [1-2] [1] Qizhe Xie, and others, "Unsupervised Data Augmentation for Consistency Training" [2] Hieu Pham, Quoc V. Le, “Semi-supervised Learning by Coaching” Post rebuttal: The reviewer appreciate the detailed response from the authors. The only part the reviewer is not quite sure about is the selection of NAO as the baseline.


Review 4

Summary and Contributions: The authors propose a semi-supervised neural architecture search algorithm called SemiNAS that promises to reduce the computational cost over other NAS methods by augmenting the set of architectures for which a full training until convergence must be performed with a set of untrained architectures which only require a prediction of their accuracy. The algorithm leverages the neural architecture optimizer described in [15] which uses a shallow LSTM to encode a discrete architecture x into a continuous representation e_x=f_e(x) whose accuracy is predicted using a predictor model f_p. Combined with a decoder f_d that allows for recovering a discrete model architecture from any continuous representation, the NAS moves the representation e_x toward that direction which increases the accuracy. The combination of {f_e f_p, f_D} form the controller. In each iteration, SemiNAS augments a set of (initially N) fully trained and evaluated architectures with M additional randomly generated architectures where the accuracy of the latter is predicted using f_p. The controller is then optimized on the combined set of N+M architectures. The refined controller then shifts the continuous representations of the top K performing architectures toward the direction of higher predicted accuracy via gradient ascent. The resulting architectures are fully trained and added to the (initial) set before running the next iteration of SemiNAS.

Strengths: The approach is fairly simple and straight forward and depends to a large extent on the ability of the predictor network to correctly assess and predict the accuracy of a given continuous architecture representation. SemiNAS seems applicable to many other NAS methods as it operates more like a meta algorithm that does not require a change in its underlying optimizer.

Weaknesses: The paper does not explore the effect of choosing different values of SemiNAS hyperparameters, namely N, M and K. Each benchmark is evaluated with a predefined choice which leaves the reader without any hint as to how stable the algorithm is with respect to difference (potentially suboptimal) choices of these parameters. How does SemiNAS compare to the top-ranked 42 methods on the NASBench-101 benchmark? Is it still competitive? With the exception of P-DARTS[4] and PC-DARTS[31], all 12 baselines reported in Table 2 on the ImageNet benchmark had been published in 2018 or before, which raises the question whether there are any more recent and more competitive baselines available. The Text-to-Speech (TTS) setting reports the mean opinion score (MOS) for the low resource setting where comparisons to existing state-of-the-art systems are difficult to make. However, in the high-resource setting, the authors neither report the intelligibility rate nor the MOS. If budget constraints were the reason, it would have been better to allocate the resources to the evaluation of Table 4 instead of Table 3. The results may be somewhat difficult to reproduce given that each experiment was conducted 500 time to report averaged results. This suggests that there is a high variance and potentially some non-determinism in each experiment so that a high number of repetitions is required to make results look more stable.

Correctness: The mean opinion score reported in line 301 of Section 5.2 for the low-resource setting should be quoted in Table 3. "Intelligibility Rate" is a rather uncommon metric as it does not capture the naturalness of the produced speech.

Clarity: The presentation of Algorithm 1 can be improved: It is somewhat difficult to discern which aspects of the controller and the predictor get modified when. The reader is required to go back and forth to Section 3.2 to fill in the missing pieces. A high level description of the method in prose is fine but the actual algorithm presented on page 4 should then use the definitions and equations defined in the previous sections. E.g., line 144 describes how a better continuous representation e'_x is obtained through gradient descent as e'_x = e_x + η ⋅∂p(e_x)/∂e_x. The algorithm on the other hand simply states that "for each architecture, [...] a better architecture [is obtained] using the controller by applying gradient ascent optimization with step size η. The decoder f_d is used as input to Algorithm 1 in line 1 but then f_d does not occur anywhere else in lines 2-12. Instead, the reader has to make the association that when ever Algorithm 1 generally references the "controller" it implicitly includes the decoder f_d. In line 192, the paper states a confidence interval but does not name the method based on which this confidence interval was estimated. Is this based on bootstrap resampling normal approximation or any other method? What do the authors mean when they state in line 188 that "the best test accuracy in this dataset is 94.32%"? Does this mean that an accuracy of 100% is impossible to achieve or does the best algorithm that the authors are aware of yields this accuracy?

Relation to Prior Work: Seems to be sufficiently captured although, given that SemiNAS is a meta algorithm that leverages on NAO, I would have liked to see it discussed for a broader spectrum of NAS methods.

Reproducibility: No

Additional Feedback: Since the review form does not provide a dedicated box where the authors response could be addressed other than amending categories such as "summary and contributions", "strengths", "weaknesses", etc. I am putting my comments here. I think my previous comments and questions are still valid to some extent based on the material presented in the paper, yet I will increase my overall recommendation as it seems that many of the points are common practice in the NAS community. I still have two comments/questions: Table 1: What is the point of reporting test regret in Table 1 when it already shows test accuracy? One of the two seems redundant if the best test accuracy on this dataset is given. Mean opinion scores around 2 are rated as poor, and TTS systems operating in that region are barely useful. While SemiNAS is shown to help finding better architectures under low resource settings, it would be more informative to learn how the method behaves for high resource settings.

[Author Response · NeurIPS 2020]

We thank all the reviewers for the valuable comments and suggestions.

**To Reviewer #1**: (**1**) [**Why semi-supervised method helps?**] We have a framework consisting of an encoder, a predictor
and a decoder, where the encoder and decoder together act as an autoencoder to learn the representation of architectures
via reconstruction task. This enables improvements when training with additional pseudo-labeled architecture-accuracy
pairs. Besides, we indeed use dropout as in NoisyStudent (the paper you mentioned) to help generalization. We will
add these discussions in the paper. (**2**) [**Training on 50 pairs**] It leads to severe performance drop.

**To Reviewer #2**: [**Why SemiNAS but not SemiNAO**] The basic idea of SemiNAS, leveraging unlabeled architectures
via the encoder-predictor-decoder framework and predicting the accuracy of candidate architectures to boost the search
process, is general and can be applied to various NAS algorithms as discussed in Section 3.3. NAO is only chosen as
a demonstration example. We also combine SemiNAS with other NAS algorithm (e.g., Regularized Evolution) and
conduct experiments in Table 1 to further verify its effectiveness. It is also easy to apply SemiNAS to RL based NAS
methods, by predicting the accuracy of an architecture as the reward. We will add such experiments in the new version.

**To Reviewer #3**: (**1**) [**Results in Table 1**] SemiNAS (RE) only uses 1000 (half of original RE uses) architecture-accuracy
pairs to achieve comparable accuracy, which is to show that SemiNAS can reduce the resources required. We also run
SemiNAS (RE) consuming 2000 pairs to compare with RE under the same number of queries, and it achieves 94.03%
test accuracy which outperforms RE. (**2**) [**Standard deviation on CIFAR-10**] Though NASBench-101 is conducted on
CIFAR-10, there exist some differences. It runs each model for 3 times and collect the 3 results to reduce the variance.
Moreover we run the experiment for 500 times suggested by the authors of NASBench-101 to further reduce the
variance. We show that even 0.1% is already a significant improvement on NASBench-101 via statistical method in line
191. More discussions on how to compare different algorithms via test regret and ranking for better interpretation are
included in lines 191-203. (**3**) [**Comparison with EfficientNet**] Thanks for the suggestion! We will add EfficietNet-B0
for a comparison. (**4**) [**Why built upon NAO?**] Please refer to our response to reviewer #2. As for one-shot search, it
usually uses weight sharing to reduce the time of training architectures as in ENAS and is orthogonal to the core search
algorithm. Our experiments on ImageNet and TTS use one-shot search with weight sharing. (**5**) [**The novelty and
comparison with other SSL works**] For the novelty, please refer to our response to Reviewer #1 and #2. The related
SSL works you mentioned focus on proposing novel SSL methods/algorithms; while our focus is to boost NAS by SSL
rather than proposing a new SSL algorithm (we can choose any SSL method) and mainly compare with other NAS
works. We will cite and adopt the mentioned SSL methods in the new version, and further explore more advanced SSL
techniques into NAS in future works. (**6**) [**Lines 130-141**] You are right! We will polish this part to make it clearer.
(**7**) [**Parameterization of encoder/ predictor/decoder**] The weights of encoder, predictor and decoder are independent
without sharing. As in line 109, the predictor ($f\_p$) is a multi-layer fully connected network with relu activation. (**8**)
[**Unlimited unlabeled architectures and extremely few labeled architectures?**] The gain from unlabeled architectures
will become saturated when the number of unlabeled architectures continuously increases, as shown in the Appendix.
We only use 100 architecture-accuracy pairs (N=100) in our experiments on ImageNet and TTS, and further reducing
labeled architectures leads to performance drop.

**To Reviewer #4**: (**1**) [**Selection of N, M, K**] We mainly study different M in the Appendix. For N and K, it is obvious
that larger values will result in better performances. Considering the resources constraints and our motivation, we do
not explore larger N and K. We mainly explore how small N can be to achieve comparable performance. We find
that N should be at least 100 and smaller N leads to severe performance drop. (**2**) [**"Top-ranked 42"**] Seems you
misunderstood Table 1. The ranking in Table 1 indicates the ranking of the discovered architecture among all the
candidate architectures in NASBench-101, rather than the ranking of specific NAS algorithm (in your comments). It
does not mean that there exist 42 other NAS algorithms that are better than SemiNAS. (**3**) [**Comparison on ImageNet
with other works.**] We follow the search space and tricks in ProxylessNAS, and mainly compare to works with the
same setting, while some other works use additional tricks/modules (e.g., swish activation, squeeze-and-excitation,
auto data augmentation) that can improve the accuracy for several points. We will add more missing related works to
Table 2 for general comparisons. (**4**) [**Experiments on TTS**] We evaluate the MOS for the robustness test experiment.
The MOS for Transformer TTS, NAO and SemiNAS are respectively 2.03, 2.22, 2.43, which shows the advantages
of SemiNAS. Note that we just use Griffin-Lim as the vocoder for quick comparison, and will use neural vocoder to
improve the MOS score in the new version of the paper. (**5**) [**Multiple runs on NASBench**] Running for 500 times is
suggested in the original NASBench-101 paper by its authors. We just follow this to fairly compare with other works.
(**6**) [**Algorithm 1**] We simplify the processes of NAO in Algorithm 1 by using simple description instead of complicated
equations to let the reader focus on the semi-supervised learning method rather than the underlying search algorithm.
We will add more details and explanations to make this part clearer and easier to understand. (**7**) [**Confidence interval
in line 192**] It is based on bootstrap resampling. (**8**) [**Best test accuracy in NASBench-101**] NASBench-101 contains
423k architectures and their evaluated test accuracy on CIFAR-10, among which the highest test accuracy is 94.32%,
which is the goal for NAS algorithms to achieve.

[Meta-Review · NeurIPS 2020]

The paper proposes an interesting semisupervised approach to neural architecture search: Using architecture accuracy prediction function to to train the controller (architecture generator), and shows that such approach yields efficiency improvements. Reviewers generally agree on simplicity of this method and good experimental evaluation. Reviewers 3, 4 point out a number of missing comparisons however many of these are addressed in the rebuttal. It would also be good to understand why this method work, since as reviewer points out, no new information is added by the evaluation network - which on the other hand makes the experimental confirmation interesting. Overall this is an interesting and simple method with good evaluation and results.